# The structure of prevacuolar compartments in *Neurospora crassa* as observed with super-resolution microscopy

Barry J. Bowman ●*

Department of Molecular, Cell and Developmental Biology, University of California, Santa Cruz, Santa Cruz, California, United States of America

* bbowman@ucsc.edu

## Abstract

The hyphal tips of *Neurospora crassa* have prevacuolar compartments (PVCs) of unusual size and shape. They appear to function as late endosomes/multivesicular bodies. PVCs are highly variable in size (1–3 microns) and exhibit rapid changes in structure. When visualized with tagged integral membrane proteins of the vacuole the PVCs appear as ring or horseshoe-shaped structures. Some soluble molecules that fill the lumen of mature spherical vacuoles do not appear in the lumen of the PVC but are seen in the ring or horseshoe-shaped structures. By using super-resolution microscopy I have achieved a better understanding of the structure of the PVCs. The PVC appears to form a pouch with an open end. The walls of the pouch are composed of small vesicles or tubules, approximately 250 nm in diameter. The shape of the PVC can change in a few seconds, caused by the apparent movement of the vesicles/tubules. In approximately 85% of the PVCs dynein and dynactin were observed as poorly defined lumps inside the pouch-shaped PVCs. Within the PVCs they were not attached to microtubules nor did they appear to be in direct contact with the vesicles and tubules that formed the PVCs. In the future, the structure and relatively large size of the *Neurospora* PVC may allow us to visualize protein-sorting events that occur in the formation of vacuoles.

## Introduction

*Neurospora crassa* is a filamentous fungus that grows as a network of long, tubular hyphae. Each tube is formed as a series of compartments separated by thick-walled septae. When actively growing, organelles move through a pore in the septum, mobilizing cytoplasm from older parts of the tubular network to the growing tips. The most metabolically active compartment is at the tip [1].

While examining the distribution of the vacuolar ATPase, the enzyme that transports protons into several types of organelles [2, 3], we found it to be highly concentrated in organelles similar in size to nuclei, but with dynamically changing shapes. These unusual organelles were confined to a region 50–200 microns from the hyphal tip [2, 4, 5].

**Data Availability Statement:** All relevant data are within the paper and its Supporting Information files.

**Funding:** This research was not funded by a grant or other agency. A statement to this effect is at the end of the manuscript.

**Competing interests:** I have no competing interest.

Subsequent investigations led us to conclude that these organelles are prevacuolar compartments (PVCs) [4]. In addition to the vacuolar ATPase they contain integral membrane proteins that function in the vacuole, such as the calcium/H+ transporter (CAX), a polyphosphate polymerase (VTC-1), and an NADPH oxidase (NOX) [4, 6]. Proteins found in the lumen of vacuoles, such as carboxypeptidase (CPY) or alkaline phosphatase (PHO-8) are not present in most of the prevacuolar compartments [4]. Rab GTPases can be good markers of organelle identity [7–9]. In many organisms Rab-7, is found in late endosomes and vacuoles/lysosomes [10–12], and in *N. crassa* it strongly colocalized with the vacuolar ATPase in the PVCs [4.5] Neither Rab-5, a marker of endosomes, nor ATG-8, a marker of autophagosomes was found in the PVC [4].

We hypothesized that this organelle is the equivalent of the late endosome/multivesicular body identified in other organisms [13–16]. However, there are important differences in its structure and dynamic behavior. In yeast and mammalian cells the late endosome/multivesicular body is small, 0.2–0.6 microns and thus hard to visualize with light microscopy [13]. The PVC of *N. crassa* is variable in size, but organelles of 1–3 μm diameter are typically observed. Although they often appear to be spherical the shape is highly variable and sometimes appears to have an open side. The shape can change on a time scale of a few seconds. Experiments with fluorescent soluble molecules suggest that there is no closed lumen within the apparent spherical organelles. Carboxy-DFFDA, which accumulates within mature vacuoles was not observed in the "lumen" of PVCs [4, 17]. A pigment produced in adenine auxotrophs [18, 19] accumulates in the lumen of mature vacuoles but in PVCs it appeared as a ring-like structure, sometimes with gaps in the ring [4, 18, 19]. These ring-like structures have also been observed in the basidiomycetes *Phanerochaete velutina* and *Pisolithus tinctorius* [20, 21].

We do not know what drives the dynamic changes in the shape of the PVC, but an important clue comes from the observation that dynein and dynactin are associated with the PVCs [4, 22, 23]. PVCs that formed ring-like structures often had clumps of dynactin in the interior space, enclosed by the ring. Some PVCs appeared to be empty.

The relatively large size of the late endosome compartment in *N. crassa*, the PVC, provides us an opportunity to see details of its structure and to observe changes over time in living cells. By using a microscope with super-resolution capability I have been able to see details in the structure of the PVCs that were not visible with conventional confocal microscopy. It revealed a structure quite different from a typical spherical organelle.

## Materials and methods

### Strains

Vma-1-dsRED has the vacuolar ATPase subunit A (locus NCU01207), tagged with a red fluorescent protein. Vma-1-GFP has the same protein tagged with a green fluorescent protein. Rab7-dsRED has the rab-7 GTPase (locus NCU06410) tagged with a red fluorescent protein. Construction of these strains is described in [4, 5]. bml-GFP has the tubulin beta subunit tagged with a green fluorescent protein and was obtained from the laboratory of Michael Freitag [24].

DIC-mCh has the dynein intermediate chain (ro-8 locus.), tagged with the mCherry fluorescent protein and was obtained from the laboratory of Michael Plamann [22]. A doubly tagged strain DIC-mCh, Rab-7-GFP was constructed by mating DIC-mCh, mating type a with Rab-7-GFP, mating type A. Progeny were screened by observation with a fluorescence microscope.

## Microscopy

To prepare samples for microscopy a drop of conidia was put on 100 mm petri plates containing Vogel's minimal medium with 2% sucrose and 2% agar. Plates were incubated at 30˚ for 16–18 h. An agar block of ~1–2 cm was cut from the growth front of the colony and placed on a coverslip with the hyphae in contact with the glass [17]. Heterokaryons with red-tagged and green-tagged proteins were made by making conidial mixtures of the two strains in ratios of 1:1, 3:1, and 1:3. The mixtures were then inoculated onto agar plates as above. In some experiments, organelles were visualized by using Oregon Green 488 carboxylic acid diacetate (carboxy-DFFDA) (Life Technologies). A 30 µl drop of a 3 mM solution was placed onto a coverslip, and an inverted agar block with mycelia was placed onto the coverslip and immediately examined with a microscope. To observe the effect of benomyl (methyl 1-butylcarbamoyl-2-benzimidazolecarbamate) we used the procedure of Riquelme [25]. Benomyl was diluted to 10 and 5 µg/ml in Vogel's medium. A drop of 15 µl was put on the coverslip.

Super-resolution images were acquired with an AxioObserver.Z1 Zeiss 880 confocal microscope equipped with an Airyscan detector and a 63x/1.4-NA Plan-Apochromat objective. GFP images were obtained with BP 420–480 + BP 495–550 filters. RFP images were obtained with BP 420–480 + LP 605 filters. Images were collected using the Airyscan fast mode with Zen Black software (Zeiss) and processed using Zen Black. The images in Fig 4 and S4 and S5 Movies were obtained with an inverted Eclipse TE2000-E spinning-disk (CSU-X1; Nikon) confocal microscope equipped with a Hamamatsu (ImageEM X2) EMCCD camera with a 63X 1.4-NA Plan APO objective. 488- and 561-nm lasers were used to excite GFP and RFP fluorophores. GFP data was collected through a BP 525/50 filter and RFP data was collected through a BP 593/40 filter.

## Results

### PVCs have variable shapes

Using the rab-7 GTPase as a marker, I observed the distribution of PVCs in the hyphal tip in a restricted region of the first hyphal compartment, starting ~25 microns from the tip and ending where the tubular vacuolar network begins (Fig 1). In planar optical sections many of the PVCs are roughly circular, some are horseshoe-shaped, and others look like short tubes. Carboxy-DFFDA, which has been shown to accumulate in tubular vacuoles and mature vacuolar compartments, was also added to the sample (Fig 1). Large spherical vacuoles rarely occur in the tip region of hyphae, but tubular vacuoles are abundant [5, 17]. As we reported previously, there appears to be a strong gradient in the concentration of carboxy-DFFDA, abundant in tubular vacuoles but nearly absent from the PVCs, especially those nearer the hyphal tip. The result is that in the merged images most of the PVCs are strongly red, but those closest to the tubular vacuoles have yellow regions (Fig 1) [4] . Accumulation of carboxy-DFFDA requires a soluble vacuolar esterase [17, 21]. The image suggests that PVCs nearest the tubular vacuoles are acquiring this soluble vacuolar enzyme (Fig 1).

### Visualization of PVCs in three dimensions

By rapidly collecting a series of planar optical sections I was able to visualize the three-dimensional structure of PVCs (S1 Movie). Ten optical slices are shown (Fig 2A). In these higher resolution images the PVC appears to be composed of small vesicles or tubules that are approximately 250 nm in diameter (Fig 2B). Although the PVC appears as a ring in some sections, there is clearly a large open gap in other sections. The optical sections were combined to form a three-dimension image (Fig 2C and S2 Movie). The overall shape resembles a pouch,

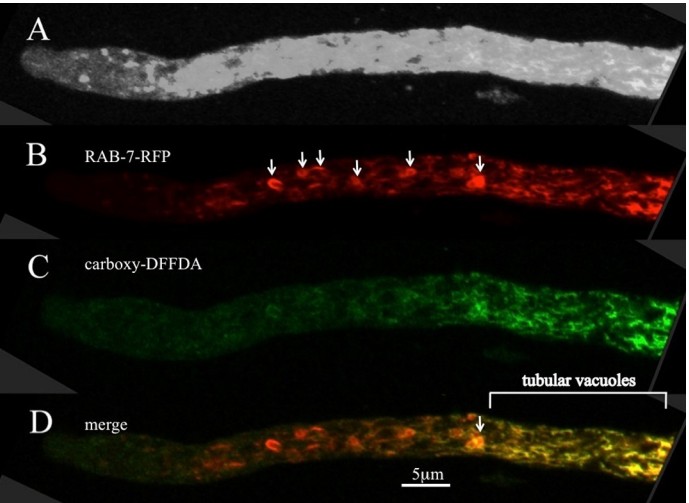

**Fig 1. Localization of RAB-7 and carboxy-DFFDA at the hyphal tip.** Panel A was intentionally overexposed to show the overall outline of the hyphal tip. Panel B shows the location of RAB-7, tagged with RFP. Arrows point to examples of PVCs. Panel C shows the location of carboxy-DFFDA, largely in the tubular vacuoles in the distal region. Panel D is the merged image for Rab-7-RFP and carboxy-DFFDA.

not a closed sphere. Imaging of a larger region of a hypha showed multiple PVCs with a pouch structure (S3 Movie). This can explain why soluble contents of the PVC, such as the pigment produced in adenine auxotrophs, do not fill the apparent lumen of the PVC [4, 19]. To put the overall size of the PVC in perspective, a spherical vesicle of the size seen in the PVC, 250 nm, may be formed by a lipid bilayer of 5 nm thickness. More than 1000 vacuolar ATPases of 12 nm diameter would fit on the surface [26].

## Rapid changes in the structure of PVCs

In live hyphae the shape of the PVC can change quickly. The vesicles that make up the PCV move rapidly compared to a small nearby vesicle that is not part of a PVC (Fig 3A). It was not possible to determine if vesicles were moving in and out of the PVC. In a longer time scale, 48 sec, the three-dimensional shape of the PVC changed significantly (Fig 3B). Some PVCs appear, in single optical sections, to be in the shape of tubes (Fig 1). By rapidly imaging single planar sections and using them to form a three-dimensional image I observed that the apparent tubes were really curved plates composed of small vesicles or tubules (Fig 3B). I observed changes over a period of 10 min using a spinning disk microscope (S4 Movie). Although the resolution was lower, this method allowed long continuous observation with little bleaching or inhibition of growth.

## Dynein and dynactin are associated with the PVC

As previously reported we visualized dynein and dynactin in living hyphae using the dynein intermediate chain tagged with mCherry and the P150 subunit of dynactin tagged with eGFP [4, 23]. By using a spinning disk confocal microscope I obtained images over a period of five min (S5 Movie). Dynein was predominantly seen near the hyphal tip. It has been shown to travel on microtubules, carrying cargo to the growth zone at the hyphal tip [27]. Dynein "comets" were observed moving rapidly towards the tip (Fig 4). The PVCs moved with the bulk flow of the cytoplasm, maintaining their position relative to the tip as the hypha elongated. A small portion of the dynein was in the shape of unstructured clumps, inside the cavity formed

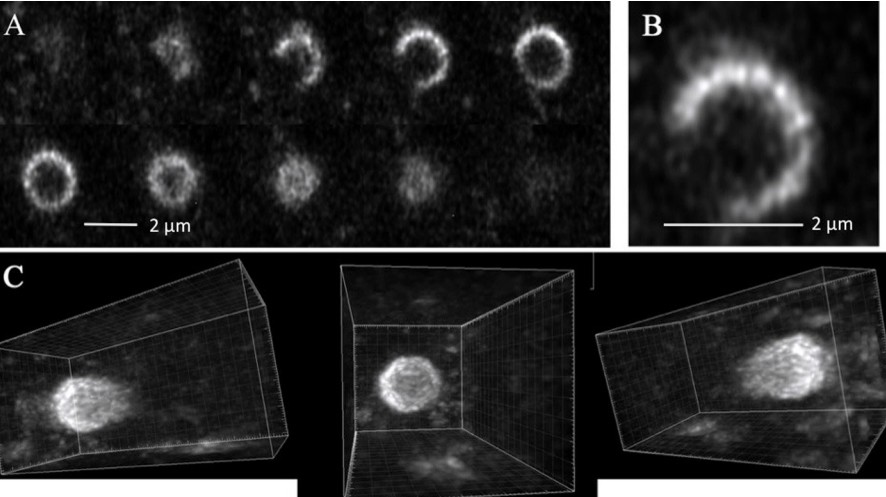

**Fig 2. Three-dimensional structure of the PVC.** The images are optical sections of a hyphal tip expressing RAB-7-RFP. 40 images were obtained over a depth of 10 microns. In panel A 10 optical sections are shown, each separated from the other by approximately 0.39 microns. Panel B shows an enlargement of the 4th optical section from panel A. Panel C shows the images as reconstructed in three dimensions, viewed from three different angles of rotation.

by the PVCs. The position of the dynein within the PVC cavity changed rapidly, with the dynein clumps moving around within the cavity of the PVC.

The structure of the dynein and its association with the PVCs was visualized at high resolution with the Zeiss/Airyscan (Fig 5). Most of the dynein, presumed to be bound to microtubules, had a linear shape, parallel to the long axis of the hypha. The dynein observed in clumps was inside the cavities of PVCs. Dynein did not completely fill the cavity of the PVCs (Figs 4 and 5 and S6 Movie).

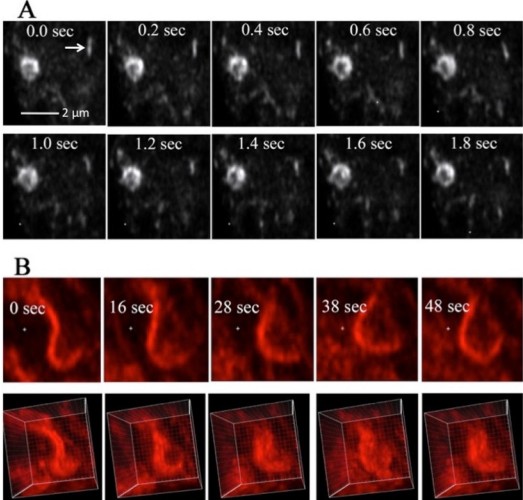

**Fig 3. Rapid changes in the shape of the PVC.** Panel A shows images of a single PVC, visualized with RAB-7-RFP, taken at intervals of 0.2 sec. For comparison, the arrows point to a small tubule which shows little change in shape or movement. In Panel B a series of optical sections was taken at each time point, over an interval of 48 sec. For each of the five images a single optical section is shown on top, and the three-dimensional reconstruction from that set of optical sections is shown below. The three-dimensional images are slightly tilted for better observation of the structure.

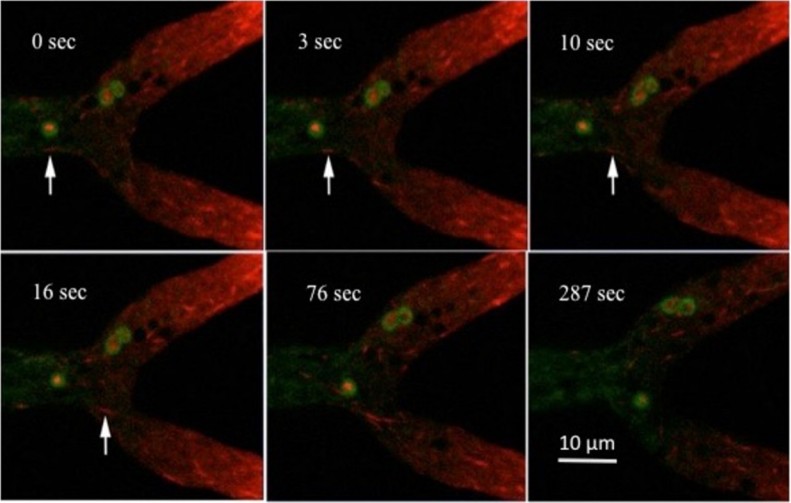

**Fig 4. Dynein is observed within the cavity of most PVCs.** Six frames from S5 Movie, obtained with a spinning disk confocal microscope, are shown. The PVCs are observed with VMA-1-GFP, and dynein is observed with DIC-mCherry. The arrows point to a dynein comet moving rapidly towards the hyphal tip.

As we previously reported [4] a minority of PVCs were not associated with observable dynein. Examination of seven hyphae showed 37 of 43 PVCs (86%) contained observable dynein.

Using tubulin tagged with GFP [24] I looked to see if microtubules were associated with the PVC or with dynein in the cavity of PVCs. No microtubules were observed within the cavity of PVCs or attached to the exterior, although it was not possible to completely rule this out, given the density of microtubules at the hyphal tip (S7 Movie). To see if maintenance of PVC structure requires microtubules, I examined the effects of benomyl, which has been shown to depolymerize tubulin in *N. crassa* [25, 28]. At concentrations as low as 5 μg/ml I observed the rapid loss of microtubule structures in the hyphal tip (Fig 6A and 6B). In the absence of benomyl most of the dynein observed in the hyphal tip is attached to microtubules within the first 10 μm of the tip. (Fig 6C-6E). After exposure to benomyl the dynein becomes distributed throughout the cytosol (Fig 6F-6N). PVCs near the hyphal tip appeared to maintain their structure (Fig 6F-6H), even in a higher concentration of benomyl (Fig 6I-6K). Because dynein was distributed throughout the cytosol, it was difficult to determine if dynein was localized within most PVCs. However, a large aggregation of dynein continued to be associated with some PVCs after exposure to benomyl (Fig 6L-6N).

## Discussion

We previously reported evidence to show that the hyphal tips of the filamentous fungus *Neurospora crassa* have organelles similar in size to nuclei, that function as prevacuolar compartments (PVCs) [4]. The PVCs contain large amounts of vacuolar ATPase and other integral membrane proteins, but are deficient in soluble vacuolar proteins such as carboxypeptidase and alkaline phosphatase. The rab-7 GTPase co-localized with the vacuolar ATPase and other integral membrane proteins of the vacuole, but the rab-5 GTPase, associated with endosomes, was not observed in PCVs. These data suggest that the PVC is the late endosome in hyphal tips.

Particularly interesting is the location of small fluorescent molecules that have been used as vacuolar markers. Carboxy-DFFDA was observed in spherical and tubular vacuoles, but was

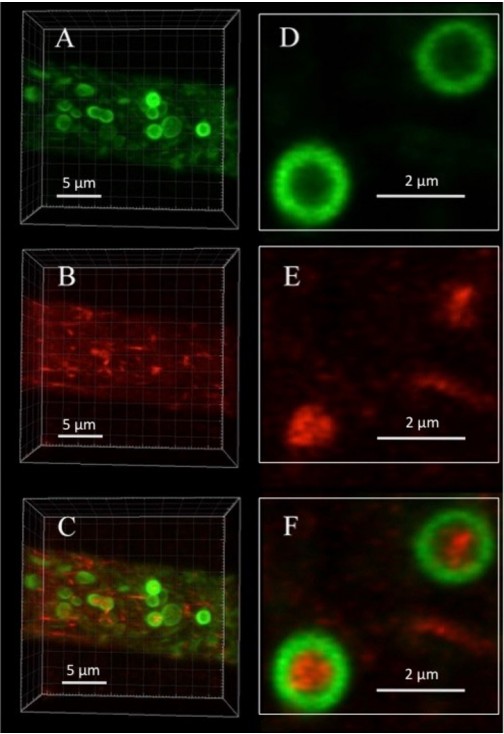

**Fig 5. High resolution imaging of dynein within PVCs.** PVCs are observed with RAB-7-GFP and dynein with DIC-mCherry. Panels A, B and C are three-dimensional reconstructions from a series of 40 optical sections obtained over a depth of 10 microns. Panels D, E and F are from a single optical section.

deficient in most, but not all, PVCs. The red pigment that accumulates in adenine auxotrophs [18, 19] was present in both PVCs and mature vacuoles. These observations are consistent with previous reports that carboxy-DFFDA accumulation depends on the activity of a soluble esterase in the vacuole, while the red pigment in adenine auxotrophs is transported by a vacuolar integral membrane protein [19, 29]. The surprising observation was that the red pigment in the PCV was not in the lumen of the PVC, but was in the shape of a ring or horseshoe or curved tubule. By using a microscope capable of producing higher resolution images, I have found that the PVC has a complex structure. It appears to be composed of small vesicles or tubules, each approximately 250 nm in diameter, which form a pouch-shaped or curved plate structure. This aggregation of vesicles is quite variable in size, ranging from 1–3 um, and undergoes dynamic changes in shape.

The interior of the pouch-shaped structures often, but not always, contain dynein and dynactin [4, 22, 23]. The dynein/dynactin has a lumpy appearance, moving rapidly within the cavity of the cup-shaped structure. Dynein/dynactin play a major role in intracellular transport by binding vesicles and moving along microtubules. The dynein/dynactin in the cavity of the PVCs was not observed to be associated with microtubules. Exposure to benomyl disrupted the structure of microtubules but did not appear to significantly change the structure of PVCs. I can only speculate that the cargo binding sites on dynein/dynactin have a role in the formation of PVCs. They do not appear to be absolutely required to maintain the structure of the PVCs because approximately 14% of PCVs had no observable dynein/dynactin.

In our previous work we observed a possible maturation process in PVCs [4]. Those PVCs nearest the tip were deficient in the soluble vacuolar marker Carboxy-DFFDA and vacuolar hydrolases. The PVCs near the tubular vacuolar network appear to have acquired these soluble

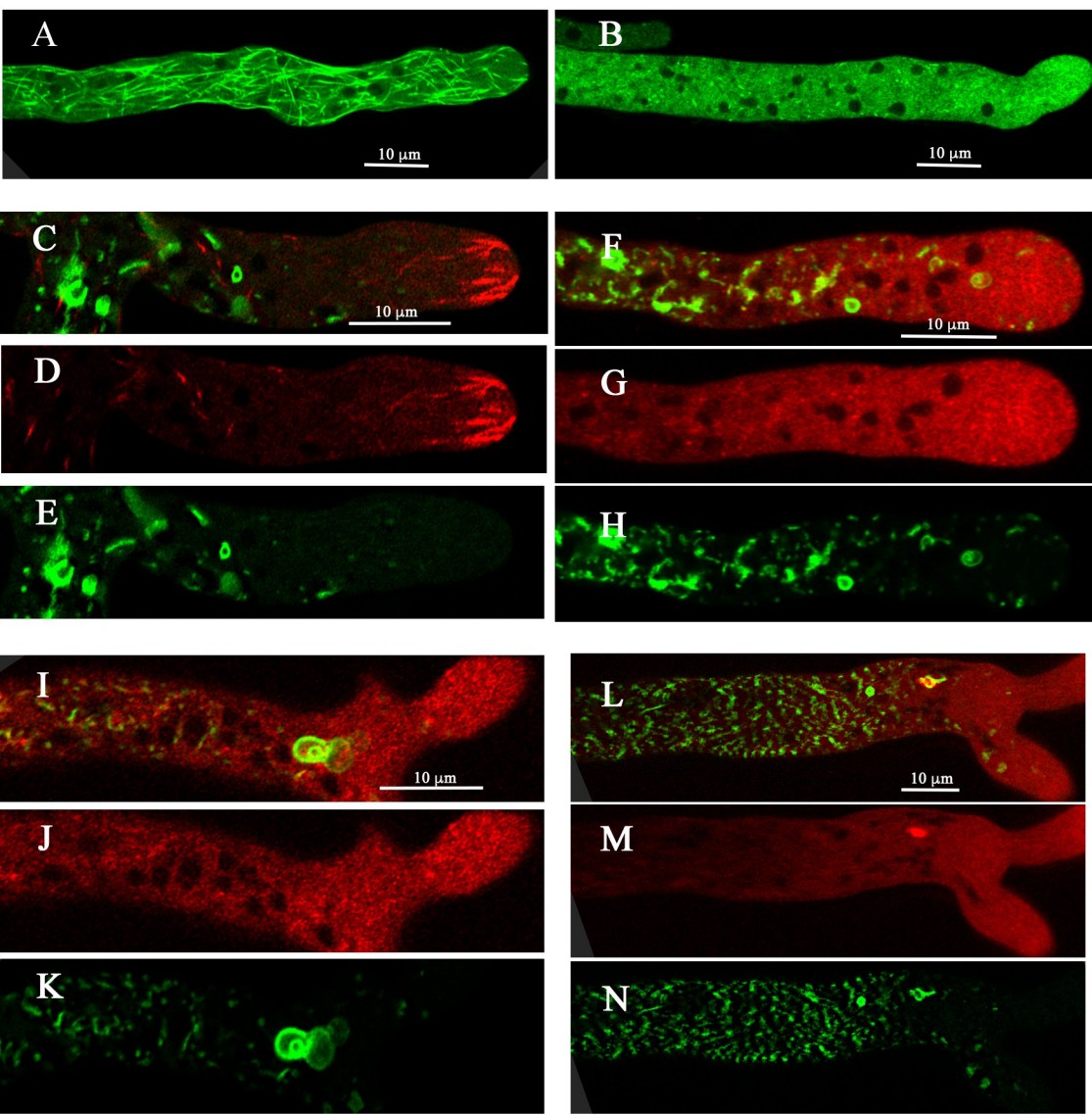

**Fig 6. Effect of microtubule depolymerization on PVC structure.** Panels A and B show the bml-GFP strain in the absence (Panel A) and presence (Panel B) of 5 µg/ml of benomyl. Panels C-N show strain DIC-mCh, Rab-7-GFP. In Panels C, D, E the hypha was not exposed to benomyl. In Panels F, G, H the hypha was exposed to 5 µg/ml benomyl. In panel I, J, K the hypha was exposed to 10 µg/ml. In Panels L, M, N, the hypha was exposed to 5 µg/ml.

contents. Individual PVCs, however, do not appear to move from the tip region towards the tubular vacuoles. They maintain their position relative the growing tip. These observations suggest the following hypothesis. Each PVC may act as a vesicle-sorting center. The dynamic changes in shape may reflect vesicles joining and leaving a PVC, a process facilitated in part by dynein/dynactin. PVCs nearest the tip gather vesicles that contain only integral membrane proteins. These vesicles may then move to other PVCs near the tubular vacuolar network where they acquire soluble vacuolar proteins.

Early endosomes, identified by association with the Rab-5 GTPase, have been investigated in filamentous fungi, with much of work being done with *Aspergillus nidulans* [7]. Less is known about the formation of vacuoles or the role of late endosomes. Thus, I don't know if the

unusually large PVCs I have observed in *N. crassa* are also present in other filamentous fungi. Similar ring-like structures were reported to be associated with tubular vacuoles in the basidiomycetes *Phanerochaete velutina* and *Pisolithus tinctorius* [20, 21]. Compared to other fungi *N. crassa* has a fast rate of growth and forms large hyphae. The large size and dynamic structural changes seen in the PVC's *of N. crassa* may allow us to observe the process of vacuole formation in real time.

## Supporting information

**S1 Movie. Optical sections of PVCs.**
(MP4)

**S2 Movie. 3D rotation of one PVC.**
(MP4)

**S3 Movie. 3D rotation of multiple PVCs.**
(MP4)

**S4 Movie. PVC shape changes in 10 min.**
(MP4)

**S5 Movie. PVCs with dynein.**
(MP4)

**S6 Movie. Rotation of PVCs with dynein.**
(MP4)

**S7 Movie. Microtubules and PVCs.**
(MP4)

## Acknowledgments

Technical support from Benjamin Abrams, UCSC Life Sciences Microscopy Center, **RRID**: SCR_021135 .

## Author Contributions

**Conceptualization:** Barry J. Bowman.

**Investigation:** Barry J. Bowman.

**Project administration:** Barry J. Bowman.

**Resources:** Barry J. Bowman.

**Writing – original draft:** Barry J. Bowman.

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
