## [Decision Letter · Decision Letter 0]

12 Dec 2022

PONE-D-22-31627The Structure of Prevacuolar Compartments in *Neurospora crassa as observed with Super-Resolution Microscopy**PLOS ONE*

Dear Dr. Bowman,

*Thank you for submitting your manuscript to PLOS ONE. After a careful review of your manuscript I concur with comments from the 3 expert reviewers that the manuscript be revised accordingly. **Therefore, I invite you to submit a revised version of the manuscript that addresses the points raised during the review process. **All the changes recommended by the 3 reviewers need to be incorporated before the acceptance of the manuscript.* *Please submit your revised manuscript by Jan 26 2023 11:59PM. If you will need more time than this to complete your revisions, please reply to this message or contact the journal office at plosone@plos.org. *

*Please include the following items when submitting your revised manuscript:*

*A rebuttal letter that responds to each point raised by the academic editor and reviewer(s). You should upload this letter as a separate file labeled 'Response to Reviewers'.*

*A marked-up copy of your manuscript that highlights changes made to the original version. You should upload this as a separate file labeled 'Revised Manuscript with Track Changes'.*

*An unmarked version of your revised paper without tracked changes. You should upload this as a separate file labeled 'Manuscript'.*

**

*We look forward to receiving your revised manuscript.*

*Kind regards,*

*Praveen Rao Juvvadi, PhD*

Academic Editor

*PLOS ONE*

“This research was not funded by a grant or other agency.  A statement to this effect is at the end of the manuscript.”

“Technical support from Benjamin Abrams, UCSC Life Sciences Microscopy Center, RRID: SCR_021135 . Purchase of the Zeiss 880 confocal microscope used in this research was made possible through the National Institutes of Health s10 Grant 1S10OD23528-01.”

“This research was not funded by a grant or other agency.  A statement to this effect is at the end of the manuscript.”

Please include your amended statements within your cover letter; we will change the online submission form on your behalf."

“I have no competing interest.”

7. In your Data Availability statement, you have not specified where the minimal data set underlying the results described in your manuscript can be found. PLOS defines a study's minimal data set as the underlying data used to reach the conclusions drawn in the manuscript and any additional data required to replicate the reported study findings in their entirety. All PLOS journals require that the minimal data set be made fully available. For more information about our data policy, please see http://journals.plos.org/plosone/s/data-availability.

*Reviewers' comments:*

*

**Comments to the Author**
*

1. Is the manuscript technically sound, and do the data support the conclusions?

*The manuscript must describe a technically sound piece of scientific research with data that supports the conclusions. Experiments must have been conducted rigorously, with appropriate controls, replication, and sample sizes. The conclusions must be drawn appropriately based on the data presented. *

*Reviewer #1: Yes*

*Reviewer #2: No*

*Reviewer #3: Yes*

*2. Has the statistical analysis been performed appropriately and rigorously? *

*Reviewer #1: N/A*

*Reviewer #2: N/A*

*Reviewer #3: Yes*

*3. Have the authors made all data underlying the findings in their manuscript fully available?*

*The PLOS Data policy requires authors to make all data underlying the findings described in their manuscript fully available without restriction, with rare exception (please refer to the Data Availability Statement in the manuscript PDF file). The data should be provided as part of the manuscript or its supporting information, or deposited to a public repository. For example, in addition to summary statistics, the data points behind means, medians and variance measures should be available. If there are restrictions on publicly sharing data—e.g. participant privacy or use of data from a third party—those must be specified.*

*Reviewer #1: Yes*

*Reviewer #2: Yes*

*Reviewer #3: Yes*

*4. Is the manuscript presented in an intelligible fashion and written in standard English?*

*PLOS ONE does not copyedit accepted manuscripts, so the language in submitted articles must be clear, correct, and unambiguous. Any typographical or grammatical errors should be corrected at revision, so please note any specific errors here.*

*Reviewer #1: Yes*

*Reviewer #2: Yes*

*Reviewer #3: Yes*

*5. Review Comments to the Author*

*Please use the space provided to explain your answers to the questions above. You may also include additional comments for the author, including concerns about dual publication, research ethics, or publication ethics. (Please upload your review as an attachment if it exceeds 20,000 characters)*

*Reviewer #1: Bowman analyzed PVCs in Neurospora crassa in detail by super-resolution microscopy using Rab7 as a marker. The results revealed a pouch-like structure, which contains dynein and dynactin. The following points need to be further examined:*

1. L237-241, if the relationship between dynein localization in PVC and microtubules is to be further investigated, microtubule polymerization inhibitors should be used to see what happens to the localization.

2. L289-290, an actual example should be provided how vesicles are entering and exiting the PVC.

Italicize: L30 Neurospora; L296,297 Aspergillus nidulans; L301 Phanerochaete

L87, 2012a].

L154, which image was overexposed shown in Panel A?

*L207, dynein intermediate chain (DIC)*

*Reviewer #2: The author describes a detailed observation of prevacuolar compartments (PVCs) in the filamentous fungus Neurospora crassa. The fungus exhibits a larger size and dynamic structures of PVCs, and thus it is a good model to characterize the organellar features. This study employs super-resolution microscopy, which revealed that the PVCs have a pouch shape with vesicles and tubules. Importantly, dynein and dynactin, components of microtubule motor, were surrounded with PVCs.*

However, the present study does not contain many new findings because similar observations were already conducted by the same author (Bowman et al., 2015. Characterization of a Novel Prevacuolar Compartment in Neurospora crassa. Eukaryot Cell 14, 1253–1263). The observations of components such as Rab7 and dynein look mostly similar to those in the previous publication, in which PVCs were extensively characterized. Although dynamic movement of PVCs is shown in the supplementary movie, figure pictures do not illustrate scientifically significant aspects.

Comments.

l. 49 Forgac, 2007 is not related to the localization in hyphae.

l. 56-57 Should Bowman et al., 2015 be cited for the result of CPY and PHO-8?

l. 164 In the picture of Fig. 2B, the size of small vesicles or tubules is not shown clearly.

*l. 238 No results on microtubules are shown.*

*Reviewer #3: The manuscript titled “The Structure of Prevacuolar Compartments in Neurospora crassa as observed with Super-Resolution Microscopy” by Barry Bowman describes very new findings about the prevacuolar compartments (PVCs) of Neurospora crassa. The manuscript is technically sound providing deeper insight to the PVC that appears to be rapidly changing in their structure.*

The manuscript may be accepted after minor revisions as suggested below.

1. This is a single author manuscript. Therefore, “we” may be replaced with “I” in the sentences like (i) “By using super-resolution microscopy we have achieved a better understanding of the structure of the PVCs (line 21-23);

(ii) “We hypothesize that…. (line 65)”; and

(iii) “We previously reported … (line 245)” etc.

2. The reference to the figures are not consistent in the manuscript. At some places it is in the main sentence and at some other places it is in the parentheses,

e. g.

line: 138: shown in figure 1..

line 151. The image in Fig 1

line 165: (Fig 2B)…

It is better to use a consistent style, preferably referring to the figures within the parentheses as in the line 165.

3. Line 283: In our previous work we observed a possible maturation process in PVCs…

Please provide supporting reference for this line.

*4. Fig. 1: Kindly insert a single space in the scale, it should be 5 μm.*

*While revising your submission, please upload your figure files to the Preflight Analysis and Conversion Engine (PACE) digital diagnostic tool, https://pacev2.apexcovantage.com/. PACE helps ensure that figures meet PLOS requirements. To use PACE, you must first register as a user. Registration is free. Then, login and navigate to the UPLOAD tab, where you will find detailed instructions on how to use the tool. If you encounter any issues or have any questions when using PACE, please email PLOS at figures@plos.org. Please note that Supporting Information files do not need this step.*

---

## [Author Response · Author response to Decision Letter 0]

24 Feb 2023

Response to Reviewers.

I thank the reviewers for carefully reading my manuscript.

Reviewer #1. I have followed through on the suggestion to use a microtubule polymerization inhibitor. The results are shown in figure 6 and described in lines 244 - 262. The results indicate that microtubules are not required for the maintenance of PVC structure and that the aggregation of dynein associated with PVCs can also be maintained when microtubules have depolymerized.

The reviewer asks for an example (I assume an image or video) of vesicles entering and exiting a PVC. I would love to see this but it has not been technically possible, at least not with the microscopes available to me. Section 3.3 in the paper is my best attempt to do so. I see rapid changes in structure but I have not observed actual fusion. Vesicles move very quickly, and the actual cargo vesicles may be as small as 90 nm, at the limit of detection for light microscopy.

In the Discussion I say “These observations suggest a hypothesis.” I put it as a possible explanation, but I don’t claim to have demonstrated it.

L30, L296, 297, 301, I italicized the names

L154 Panel A in the figure was intentionally overexposed to make the outline of the tip more visible.

Reviewer #2. The reviewer feels that the manuscript does add scientifically significant aspects beyond what was presented in my 2015 paper. Part of the motivation for doing this high-resolution microscopy was to address questions raised by reviewers of that 2015 paper. In that paper I suggested that there was no “inside” to the PVC. With soluble internal markers the PVCs often appeared as donuts, not filled spheres. I was requested to deemphasize this conclusion because there was not sufficient visual evidence. The structural detail in my current manuscript provides a much more accurate understanding of the structure of this organelle.

L49 Forgac reference removed 

L56-57 Citation of Bowman 2015 added

L164 The microscopy does not produce a clear edge for the vesicles. In the figure I show a 2 micron scale bar. The vesicles vary in size but their diameters are approximately 1/8th the length of that line.

L238 The images for microtubules are in the supplementary movie. I have added Fig 6 to give more information about the role of microtubules.

Reviewer #3. This is the only single authored manuscript in my career, and it seems a little odd to say “I observed…” But the reviewer is correct. I changed “we” to “I”, except when referring to previous multi-authored papers.

To be consistent, reference to Figs are now in parentheses e.g. (Fig. 1).

L283 the citation has been added.

Fig 1 micron scale fixed.

---

## [Editor Report · Decision Letter 1]

1 Mar 2023

The Structure of Prevacuolar Compartments in *Neurospora crassa as observed with Super-Resolution Microscopy*

*PONE-D-22-31627R1*

*Dear Dr. Bowman,*

*We’re pleased to inform you that your manuscript has been judged scientifically suitable for publication and will be formally accepted for publication once it meets all outstanding technical requirements.*

*Within one week, you’ll receive an e-mail detailing the required amendments. When these have been addressed, you’ll receive a formal acceptance letter and your manuscript will be scheduled for publication.*

*An invoice for payment will follow shortly after the formal acceptance. To ensure an efficient process, please log into Editorial Manager at http://www.editorialmanager.com/pone/, click the 'Update My Information' link at the top of the page, and double check that your user information is up-to-date. If you have any billing related questions, please contact our Author Billing department directly at authorbilling@plos.org.*

*If your institution or institutions have a press office, please notify them about your upcoming paper to help maximize its impact. If they’ll be preparing press materials, please inform our press team as soon as possible -- no later than 48 hours after receiving the formal acceptance. Your manuscript will remain under strict press embargo until 2 pm Eastern Time on the date of publication. For more information, please contact onepress@plos.org.*

*Kind regards,*

*Praveen Rao Juvvadi, PhD*

Academic Editor

*PLOS ONE*

* *

*Additional Editor Comments (optional):*

*Dear Dr. Bowman,*

Thank you for addressing the comments raised by the three reviewers. I believe the manuscript has been sufficiently improved based on their suggestions.

I do not have any further comments.

Best,

*Praveen*

* *
---

## [Editor Report · Acceptance letter]

6 Mar 2023

PONE-D-22-31627R1 

The Structure of Prevacuolar Compartments in *Neurospora crassa * as observed with Super-Resolution Microscopy 

Dear Dr. Bowman:

I'm pleased to inform you that your manuscript has been deemed suitable for publication in PLOS ONE. Congratulations! Your manuscript is now with our production department. 

Kind regards, 

on behalf of

Dr. Praveen Rao Juvvadi 

Academic Editor

PLOS ONE